# Astronomical Investigation to Verify the Calendar Theory of the Nasca Lines

Christiane Richter [1] , Bernd Teichert [1,*] and Karel Pavelka [2]

1 Faculty of Spatial Information, University of Applied Sciences (HTW), Friedrich-List-Platz 1, 01069 Dresden, Germany; christiane.richter@htw-dresden.de

2 Faculty of Civil Engineering, Czech Technical University in Prague, Thakurova 7, 16629 Prague, Czech Republic; pavelka@fsv.cvut.cz

* Correspondence: bernd.teichert@htw-dresden.de

**Abstract:** As in many regions of the world, astronomy also played a major role in the ancient Peruvian cultures. However, the discussion of the astronomical relevance of the Nasca geoglyphs is very controversial. A really precise and extensive investigation of astronomical phenomena has not yet taken place; the necessary data were simply missing. In the Nasca project Dresden, these data have been recorded in recent years and stored in an Oracle database. In the very first step, all geoglyphs with an astronomical orientation documented by Maria Reiche were checked and verified. Subsequently, all lines of the entity "straight line" were systematically examined with regard to the celestial bodies of the Sun and bright stars. For this purpose, on the one hand, the ellipsoidal azimuths of all straight lines were calculated and, on the other hand, the elevation angles in relation to the horizon with the help of digital terrain models (DTM) were determined. Corrections for refraction, the curvature of the Earth, visibility and atmospheric disturbances were largely considered. The azimuths of the celestial bodies during the Nasca period were calculated with software developed in-house (theses by students) and compared with those of the lines. As a result, it was possible to establish that there are individual straight lines that are aligned with the Sun and the seven randomly selected bright stars. However, the number of hits found does not justify the theory that the Nasca Pampas are an astronomical calendar system.

**Keywords:** astronomy; Nasca geoglyphs; Sun solstice; equinox; pre-Columbian cultures

## 1. Introduction

The coastal area in western South America from Patagonia to Ecuador is one of the driest regions on Earth. The cold Humboldt Current in the Pacific prevents cloud formation over the ocean, which means that there is almost no precipitation in this subtropical coastal zone. The only basis for human life is the valley oases of the rivers, which, during the rainy season in the Andes, lead water towards the Pacific for a few weeks per year. In these river oases on the coast of South America, several pre-Columbian cultures developed over the centuries.

Nowadays, most people only know the Inca Empire. Although the Inca Empire was the largest in terms of territory, it only existed for a relatively short time before it was conquered by the Spaniards in 1532. However, there were already many important pre-Columbian cultures before the Incas which created impressive structures and cultural assets.

One of these cultural heritage sites is located around 450 km south of the Peruvian capital Lima, in the middle of the coastal desert (compare Figure 1)—the geoglyphs at the Pampa of Nasca and Palpa.

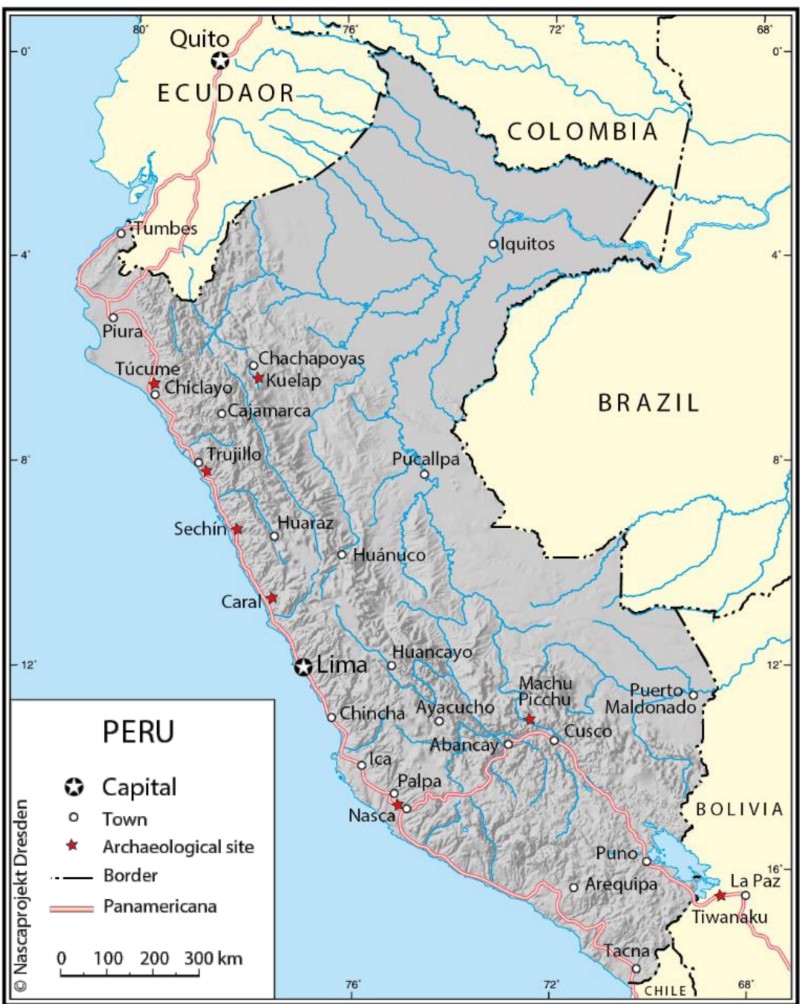

**Figure 1.** Peru—overview map (Reprinted with permission from ref. [1]. 2006 Schmid).

The geoglyphs are some of the most fascinating and mysterious archaeological sites in the world. Several hundred square kilometers of rock-strewn desert are covered with thousands of lines, large biomorphic figures and various geometric shapes (Figure 2). The drawings vary in complexity. Hundreds of them are simple straight lines, running up to twelve kilometers straight through the desert. However, there are also meandering and zigzagging lines. There are huge areas—mostly triangles and trapezoids—that look like landing strips. Lines and areas often meet in so-called line centers, sometimes situated on slightly elevated places. Between all these lines and areas, numerous figures can be found. There are anthropomorphic figures, zoomorphic designs (such as birds, fishes, llamas, a spider or a monkey), plants (such as trees or flowers) and geometric shapes, e.g., different kinds of spirals. Some of the figures measure more than 300 m in length. That is why the beauty and magnitude of the giant figures can be recognized only from the air [2].

These geoglyphs represent an artistry of a special kind, neither carved nor painted. They were caused by removing the darker, oxidized layer of the desert's surface to expose the lighter, unoxidized sub-soil below. However, the geoglyphs certainly did not arise all at once; many of them overlap, with the oldest drawings being almost invisible. Most of these geoglyphs have their seeds in the pre-Columbian Nasca culture (approximately 200 BC–650 AC). Only a few, predominantly anthropomorphic representations have their origin in the older Paracas culture (800–200 BC) [3].

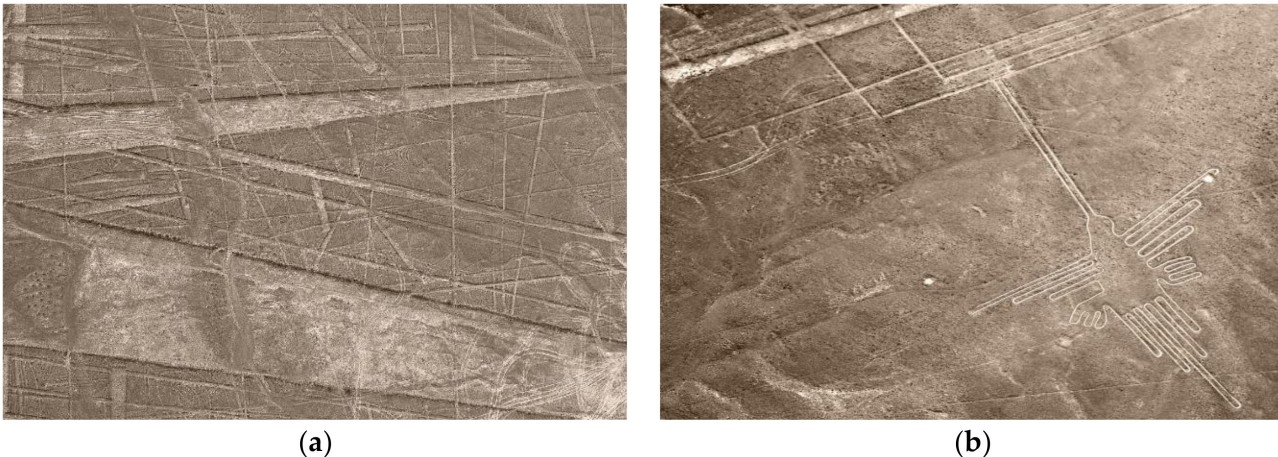

(**a**) (**b**)

**Figure 2.** Geoglyphs in the Pampa of Nasca: (**a**) lines and areas, (**b**) figure of the hummingbird.

The fact that these mysterious drawings have remained intact for so many centuries is due to the geographic and climatic situation on the Peruvian coast. The so-far unsolved question is why the people of the Nasca period created these drawings. The mysteries surrounding the Nasca geoglyphs are almost as fascinating as the spectacular drawings themselves. When Dr. Paul Kosok, professor of history at the Long Island University, saw the Sun setting exactly over a narrow line in the desert of Nasca on 21 December 1941 (see Figure 3), the day of the summer solstice, he concluded that the Pampa of Nasca is "The Largest Astronomy Book in the World". The Dresden-born teacher Dr. Maria Reiche followed his ideas, and she found many lines, areas and figures that provide evidence for an astronomical orientation. As Maria Reiche researched the lines for more than 40 years, she became known as the Lady of the Lines [4].

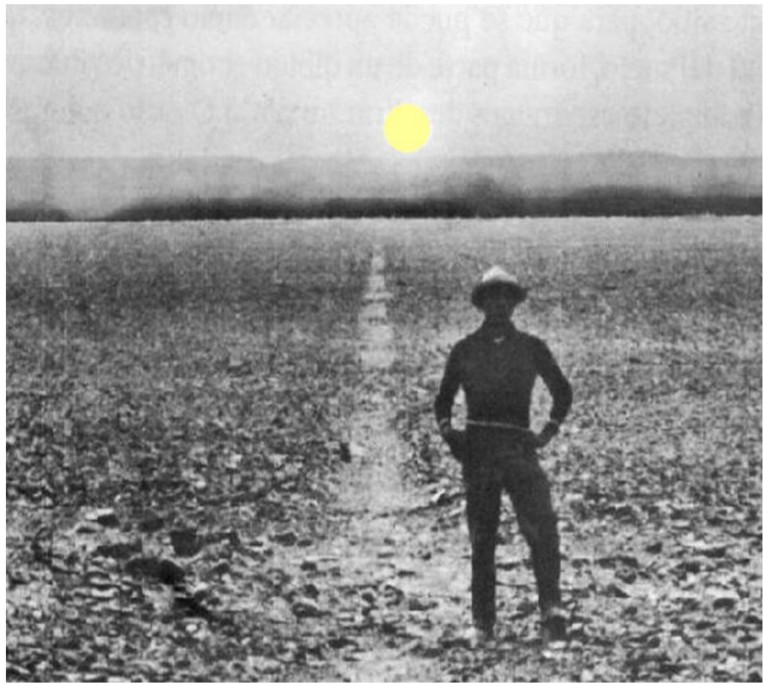

**Figure 3.** Astronomical orientation of one of the lines—Sun setting on 21 December 1941 (Reprinted with permission from ref. [4], 1989 Reiche).

The first seemingly systematic astronomical approach was conducted by Gerald Hawkins in 1969. He investigated only a few lines within a very small area compared to

the relevant pampa and concluded that the Nasca Lines, as a whole, cannot be explained as astronomical, nor are they calendric. However, because of his special method of investigation, these negative results cannot be regarded as the final word on any astronomical hypothesis for the origin of the lines. In more recent archaeological investigations in the Palpa area, the astronomical theory is completely negated again [5].

Today, it seems to be established that the lines once served as ritual places for religious ceremonies. However, the nature of these ceremonies and the precise purpose they served have still not been resolved. It is certain that the shamans of most of the ancient cultures had astronomical knowledge, which definitely had an influence on religious ceremonies and rituals. Therefore, it is not astonishing that there are correlations between some geoglyphs and celestial bodies. Therefore, the verification of this astronomical theory is one special task of the Nasca project at the University of Applied Sciences in Dresden, Germany [3].

## 2. The Sun Worship in Pre-Columbian Cultures on the Current Territory of Peru

### 2.1. Astronomical Background

There is evidence that most of the cultures on the territory of today's Peru worshiped the Sun. In this context, the alignment of structures to solstice events and equinoxes plays a major role because both solstices and equinoxes have been celebrated by many cultures throughout human history.

Two solstices occur annually around 21 June and 21 December and these are the times of the astronomical beginning of summer and winter. Equinoxes are the calendar days on which the Sun crosses the celestial equator and therefore day and night are about the same length. The equinoxes are the times of the astronomical beginning of spring or autumn. However, when we talk about summer and winter solstices, we have to consider that these terms are ambiguous, since summer in the Northern Hemisphere is winter in the Southern Hemisphere and vice versa. The same applies to the spring and autumn equinoxes. Therefore, in the following, we use the terms December or June solstice and March or September equinox.

A solstice is an event that occurs when the Sun reaches its greatest north or south declination in the course of a solar year. The declination δ is one of the two angles that locate a point on the celestial sphere in the so-called equatorial coordinate system (Figure 4a). It is measured perpendicular, north or south, to the celestial equator. The other one is the right ascension α. It is the angular distance measured eastward along the celestial equator from the vernal equinox to the point in question above the Earth.

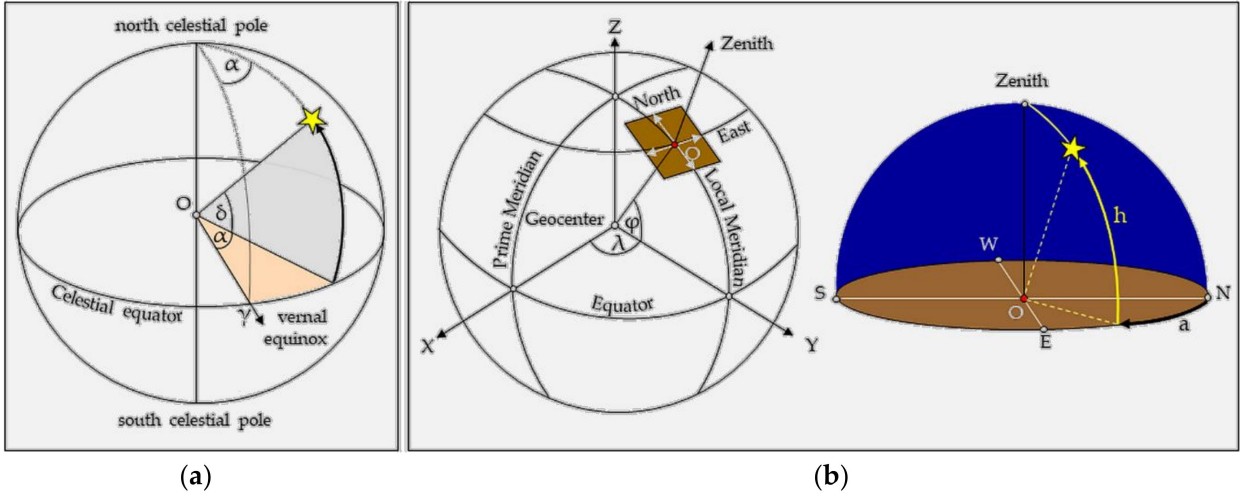

|  (a)  |  (b)  |

**Figure 4.** (**a**) Equatorial coordinate system: right ascension α and declination δ as seen from outside the celestial sphere; (**b**) horizontal system: the coordinates of a celestial body are expressed in terms of the elevation angle h and azimuth a, which depend on the position of the observer O on Earth.

An astronomical coordinate system that is frequently used to observe celestial bodies from the Earth is the local horizontal system (see Figure 4b) because this is based on the position of the observer on Earth and uses the observer's local horizon as the fundamental plane. The coordinates of a celestial body are expressed in terms of the altitude (or elevation angle) h and azimuth a. The altitude h is the angle between the object and the observer's local horizon, while the azimuth a is the angle usually measured clockwise from true north around the observer's horizon.

The determination of the declination δ of any object is given by the well-known formula

$$\sin \delta = (\sin \varphi \sin h + \cos \varphi \cos h \cos a) \tag{1}$$

For an altitude h = 0° (Horizon), it follows

$$\sin \delta = \cos \varphi \cos a \tag{2}$$

Hence, the astronomical azimuth of the sunrise or sunset on the ideal horizon (altitude h = 0°) is obtained from a simple equation:

$$a = arccos\left(\frac{\sin \delta}{\cos \varphi}\right) \tag{3}$$

where the declination of the Sun is −23.5° < δ < +23.5° and φ is the latitude of the position of the observer. The maximum declination of 23.5 degrees corresponds to the solstice in June; the minimum −23.5 degrees is reached at the solstice in December. With a declination of 0°, we speak of the equinox.

For Peru, the mean latitude is −9.189967 degrees and thus the approximate azimuth of the sunrise during the June solstice is around 66 degrees, while the sunset is at 294 degrees. On the other hand, during the December solstice, the sunrise is at approximately 114 degrees, while the sunset is at 246 degrees. The azimuth of the sunrise on the equinoxes is 90 degrees, and it is 270 degrees for the sunset.

Both solstice events and equinoxes played an important role in pre-Columbian cultures. This is particularly reflected in their buildings and structures, as the following examples are intended to show.

### 2.2. The Sun Worship of the Incas

The best example of the importance of the Sun in pre-Columbian cultures are the Incas (1200–1572). The Incas worshiped the Sun as the supreme god. It is common knowledge that the Incas considered their king to be the "son of the Sun". It is therefore not surprising that many shrines (in Quechua: huacas) were associated with solar worship and many structures were oriented to the Sun. It is interesting how much the Incas focused on the solar astronomy and how many structures with astronomical significance can be found throughout the Inca Empire. Some examples are documented, e.g., in [6].

Machu Picchu, the world-famous temple complex in the sacred valley in the Peruvian Andes, should be mentioned as only one example. An explanation of the astronomical orientation of some buildings, e.g., the Temple of the Sun (Torreón) and the Temple of the Three Windows, is provided, among others, in [6,7]. Therefore, the Temple of the Three Windows, e.g., faces towards the direction of the sunset at December solstice, while the window in the major shrine Torreón is open to the direction of the sunrise at June solstice. Moreover, it is also appropriate to name the "Intihuatana"—a hewn boulder associated with the astronomic clock or calendar of the Inca (Figure 5). Its name is derived from the Quechua language and is literally an instrument or place to "tie up the Sun". Such ritual stones can be found in many places and different cultures in the area of today's Peru.

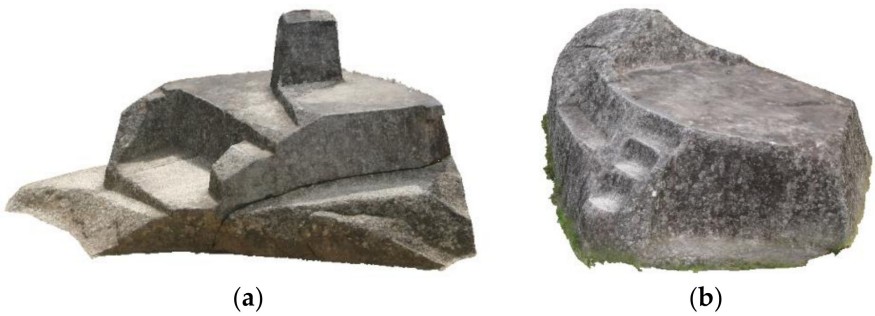

<div style="text-align:center">(<b>a</b>)　　　　　　　　　　　　　　　　　(<b>b</b>)</div>

**Figure 5.** Digital models of the well-known objects in Machu Picchu: (**a**) Intihuatana, often referred to as an astronomical observatory, (**b**) "Altar", ritual stone.

However, the sun cult played a role long before the Incas built their empire. Thus, the Sun influenced the construction of temples and structures in most of the pre-Columbian cultures. Some of these are presented as examples in the following chapter.

### 2.3. The Sun Worship in the Pre-Inca Cultures

#### 2.3.1. Norte Chico/Caral

The astronomical alignment of temples and buildings can be observed in almost all pre-Columbian cultures in the area of today's Peru; e.g., in Norte Chico on the north-central coast of Peru, about 200 km north of Lima, where a cluster of about 18 urban settlements can be found. The largest one is the Sacred City of Caral (2600–1800 BC), in the Supe Valley, impressive in terms of its complex and monumental architecture, including six large pyramidal structures, an extensive residential complex and five ceremonial plazas [8]. According to Malville [9] and Marroquin [10], the face of the main pyramid is oriented to the December solstice sunrise and to the June solstice sunset (Figure 6). Astronomical orientations can also be detected for the other pyramids [10].

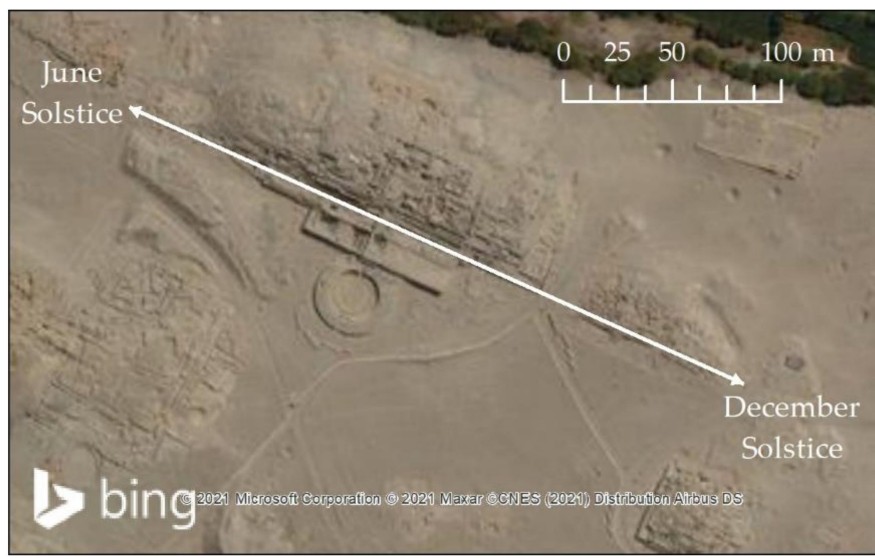

**Figure 6.** The face of the main pyramid in Caral is oriented to the sunrise at December solstice and the sunset at June solstice (source: Bing Maps © 2010 Microsoft Corporation and its data suppliers).

#### 2.3.2. Casma/Sechín

One of the oldest pre-Columbian cultures, the Casma/Sechin culture (about 3500–200 BC), refers to a large concentration of archaeological sites in the north coast of Peru, about 350 km north of the present-day capital Lima. The sites are situated in the Casma-Sechín River Basin, close to the Pacific coast.

One of the archaeological sites of the Casma/Sechín culture is a complex that includes four main local sites: Sechín Bajo, Taukachi-Konkán, Sechín Alto and the Temple of Sechín Mountain (Templo de Cerro Sechín). The most recent investigations at the site of Sechín Bajo resulted in the discovery of a sequence of three building complexes built between 3500 and 1500 BC [11]. That means that the Casma/Sechin culture is now considered the oldest civilization of the Americas. The Sechín Alto complex (1800–900 BC) is the largest structure of the Casma/Sechín Valley and it consists of several constructions. Slightly northwest of the site of Sechin Alto, the Taukachi-Konkán (approximately 2100–1000 BC) site can be found. According to Malville [9,12], the average orientation of all three complexes is 65.3°, close to the sunrise at the June solstice.

The ceremonial center Chankillo is located about 10 km southeast of Sechín and it is dated to around 300 BC [13]. The site consists of multiple structures, plazas and courtyards, spread over an area of approximately 4 km². The most impressive part of this site is a fortified temple on top of a hill. Towards the east of the temple is a large area with several buildings, a huge plaza and another very impressive structure called the Thirteen Towers. It runs north to south along a low nearby ridge.

Both structures, the Temple and the Thirteen Towers, form an artificial "toothed" horizon when viewed from the ground. According to Ghezzi and Ruggles [13], one can observe the rising and setting Sun over the course of the year from two observation points to the west and east of the towers (Figure 7). The Chankillo towers thus provide evidence of early solar horizon observations and of the existence of sophisticated sun cults.

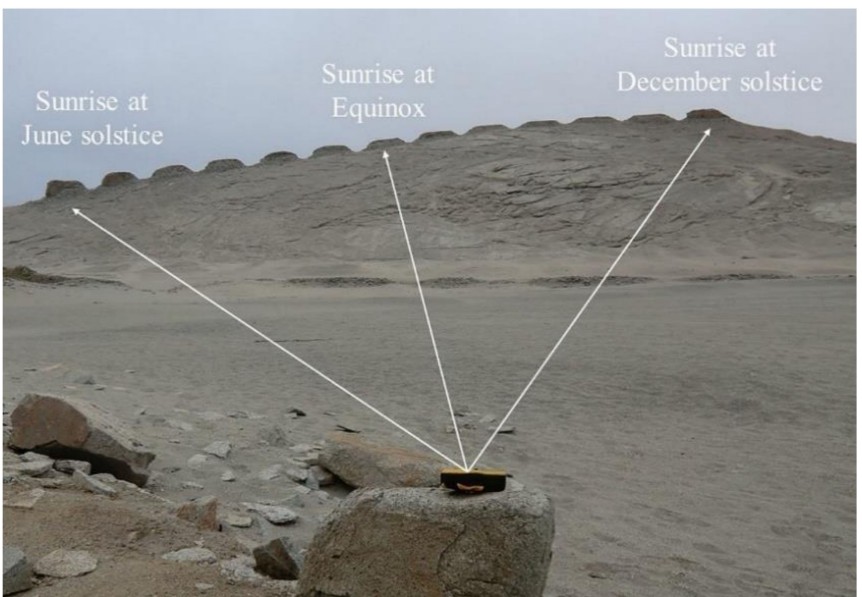

**Figure 7.** According to Ghezzi and Rugless [13], the Thirteen Towers in Chankillo as viewed from the western observation point annotated with the positions of sunrise at the solstices and the temporal equinoxes (photo: Ch. Richter, 2011).

2.3.3. Tiwanaku

Another example that should not be unmentioned is the Tiwanaku (1500 BC–1000 AC) culture in the Andes. At an altitude of about 4000 m around Lake Titicaca, they were an influential forerunner of the Inca. The Tiwanaku ceremonial and pilgrimage center is located about 15 km southeast of Lake Titicaca in Bolivia. One of the most imposing structures in Tiwanaku is the Akapana Pyramid, a giant step pyramid. Besides the Akapana Pyramid are two other major ceremonial structures of the city, the Semisubterranean Temple and another building known as the Palacio. A very impressive feature is a rectangular enclosure, known as Kalasasasaya, which consists of alternating high stone pillars and smaller rectangular blocks. The entire structure is oriented towards the two equinoxes. In

the center of the Kalasasaya, there is a large stone which is the original observation point. From this point, you could see the Sun descending over the pillars every evening. At the equinox, the Sun descends over the central pillar. At the December solstice, the Sun sets over the extreme south pillar, while the sunset at the June solstice is over the extreme north pillar (Figure 8) [14].

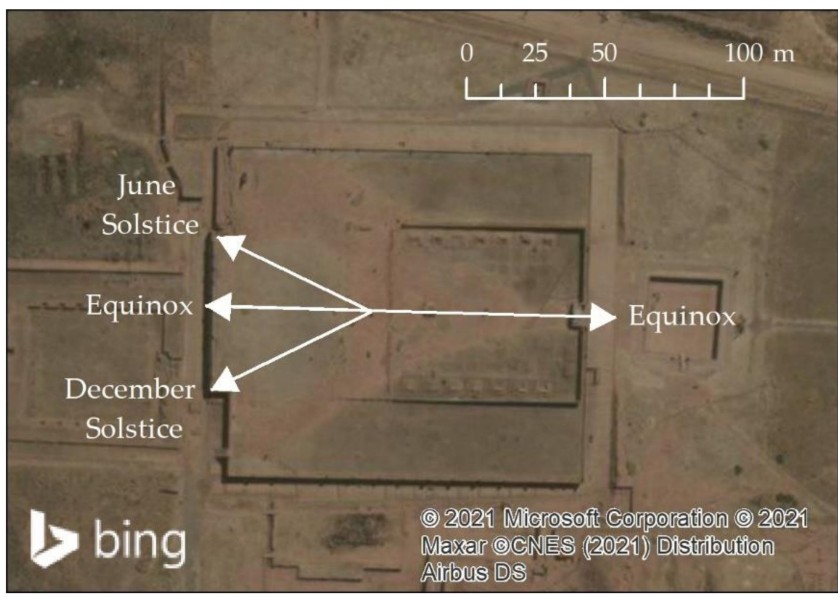

**Figure 8.** Kalasasaya with the western markers of a solar calendar, according to Allen [14] (source: Bing Maps © 2010 Microsoft Corporation and its data suppliers).

### 2.3.4. Chincha

However, there are not only buildings or structures that have an astronomical orientation. Recent archaeological research on the south coast of Peru in the central Chincha Valley examined a geoglyph complex from the Late Paracas culture (approximately 400–100 BC). The complex consists of straight lines, rock features, ceremonial mounds and settlements spread over an area of about 40 km$^2$. In Chincha, it has been proven that many lines, platform mounds and walls point to the June solstice sunset at an orientation of 294 degrees. These results were confirmed through direct field observations during the June solstice [15].

Since the Nasca culture emerged from the Paracas culture, it is reasonable to assume that some of the Nasca Lines are also oriented to the Sun.

### 3. Astronomical Investigations in the Nasca Project

As already explained, the astronomical interpretation of the Nasca Lines began with Paul Kosok and Maria Reiche in December 1941. During her 40 years of research in Nasca, Maria Reiche identified many lines, areas and figures which have an astronomical relationship to various celestial bodies (Sun, Moon, stars). Unfortunately, there are no comprehensive scientific publications on the results of Maria Reiche. She only recorded measurements and calculations in diaries and sketches but did not publish them.

Although the astronomical theory is controversial, it is undeniable that there are correlations between some lines and celestial bodies. Therefore, the verification of this theory—in continuation of the work of Maria Reiche—is one special task of the Nasca project at the University of Applied Sciences (HTW) in Dresden, Germany.

Based on the astronomical idea of Maria Reiche, there are two objectives for the astronomical approach. The first and major task is the examination of the correlation between the lines and the rising or setting of celestial bodies. Another long-term objective of the astronomical approach is the examination of the conformity of the celestial constellations of stars with some Nasca figures. This task is not part of the following considerations.

### 3.1. Requirements for the Astronomical Investigations

In order to prove whether the ancient lines in the desert near Nasca show a preference for the directions of the Sun, the Moon, planets or brighter stars, a fully automatic procedure is necessary. The very first step for this research was the geodetic data acquisition of all the lines, line centers, areas, figures and remarkable point objects (e.g., stone heaps) at a reliable accuracy. These data have been captured over the years using highly accurate measurement methods, e.g., GNSS, but also by using aerial photos and satellite images. All these data are stored in an Oracle database which is the basis of the so-called NascaGIS (see, e.g., [16]).

Another important prerequisite for the astronomical approach is a digital elevation model (DEM) because the astronomical altitude or elevation angle depends on the surrounding landscape. Therefore, the results of the Shuttle Radar Topography Mission (SRTM) as well as a DEM generated from ASTER satellite data are used [17].

By definition, the plane of the astronomical horizon is perpendicular to the plumb line at the observer's point. Mountains, valleys or buildings are in the observed direction, so this theoretical astronomical horizon deviates by the elevation angle h, which can be several degrees from the actual horizon (see Figure 9). A star rises at point $P_0$ with the azimuth $\alpha_0$; however, it only becomes visible at point $P_H$ with the azimuth $\alpha_H$ (without refraction effects). The same applies to setting stars. Therefore, there has to be a correction of the azimuth, depending on the elevation angle.

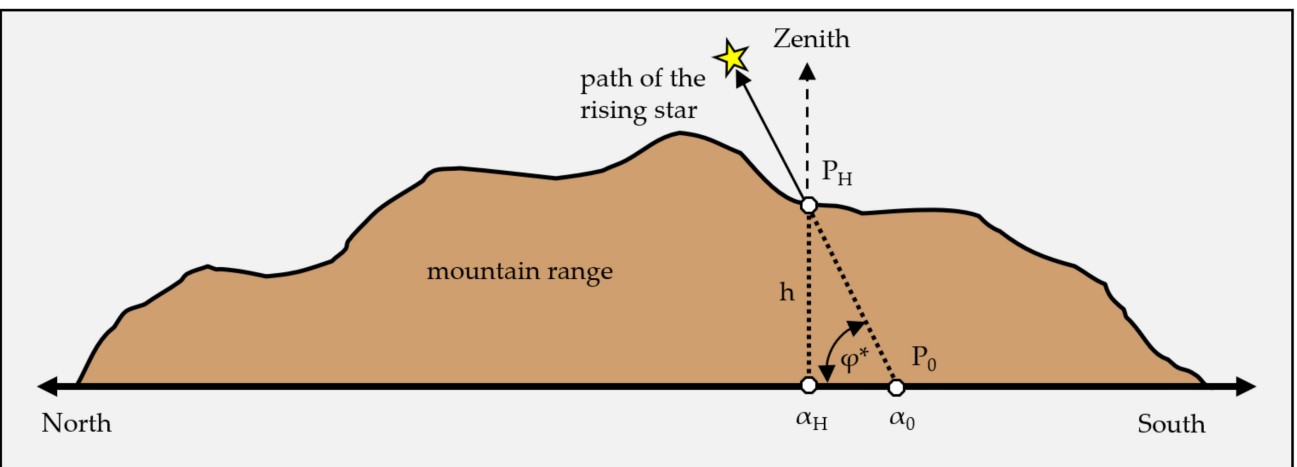

**Figure 9.** Change in azimuth and elevation angle depending on the terrain [17].

According to Aveni [18] (p. 107), for elevation angles less than 5 degrees, azimuth correction with sufficient accuracy is obtained according to Equation (4), where h is the elevation angle above the astronomical horizon and $\varphi_0$ is the geographical latitude at the observer's point $P_0$ (see Figure 9).

$$\Delta\alpha = \frac{h}{\tan\varphi^*} \text{ with } \varphi^* = 90° - \varphi_0 \tag{4}$$

For the Nasca region ($\varphi_0 \approx 15°$), the corrections in the azimuth of the stars are shown in Table 1. Corrections in the azimuth for the corresponding elevation angle must therefore be observed.

**Table 1.** Corrections in the azimuth for the corresponding elevation angle.

| h | 1° | 2° | 3° | 4° | 5° |
|---|---|---|---|---|---|
| $\Delta\alpha$ | 0°16′ | 0°32′ | 0°48′ | 1°04′ | 1°20′ |

Depending on whether the stars are rising or setting, the azimuth at point $P_H$ is derived from the following equation:

$$\text{For rising stars:} \quad \alpha_H = \alpha_0 - \Delta\alpha$$
$$\text{For setting stars:} \quad \alpha_H = \alpha_0 + \Delta\alpha \tag{5}$$

It should be noted that the equations refer to the Southern Hemisphere; the signs would be in the Northern Hemisphere and vice versa.

Further azimuth corrections must be considered because of the atmospheric effects of absorption and extinction, since the visibility on the horizon depends on the atmospheric conditions such as pollution, haze and clouds. According to local people, visibility near the horizon has decreased enormously in the past decades.

According to investigations by [19] (p. 161), the elevation angle close to the horizon for the first and last visibility of a star is roughly the same as its brightness. The brightness of a star is given by a special system of stellar magnitudes. Faintest stars have the highest value and the brightest stars the lowest. Polaris, the North Pole Star, is of Magnitude 2, while the faintest star visible to the naked eye under very good conditions is of Magnitude 6. There are about 50 stars brighter than Magnitude 2. Stars brighter than this have negative magnitudes; Sirius, the brightest star in the sky, is of Magnitude −1.45, while the Full Moon and the Sun have magnitudes of −12.6 and −26.7, respectively.

That would mean that very bright objects such as the Sun, Moon, Venus, Jupiter and Sirius with a negative brightness (magnitude < 0) are visible down to the horizon. Stars with a brightness of 1 would be visible up to an elevation angle of 1° above the horizon and so on. Since the opinions of the archaeoastronomers differ here, an investigation of this effect would be appropriate. To what extent these results can then be transferred to the conditions during the Nasca period remains questionable.

Once the accurate geometry of the lines and the DEM are stored in the database of the NascaGIS, the astronomical azimuth and the altitude can be computed. Within the scope of many diploma theses (e.g., [20–23]) at the University of Applied Sciences Dresden, several software programs were developed so that the correlations between the lines and the positions of stars, planets, the Sun and the Moon can be calculated during the entire Nasca time.

### 3.2. Verification of Maria Reiche's Results

As already mentioned, Maria Reiche identified many lines, areas and figures with an astronomical orientation. Therefore, the first task in the Nasca project was to verify these results. Therefore, in the first step, relevant geoglyphs were identified from Maria Reiche's publications and documents and their astronomical orientation was checked. Some examples of striking lines and figures with their astronomically relevant celestial bodies are briefly presented below [24–26].

Figure 10 shows several lines and stone heaps with an astronomical orientation in the main area of the Nasca Pampa, west of the Pan-American Highway, according to Maria Reiche [24]. A bird figure, the so-called condor, can be seen in the lower left area. The two highlighted lines running through the figure are aligned with the sunset at the December solstice and to the June solstice. In the middle of the picture, northeast of the condor, there is an approximately 850 × 110 m rectangle with three large piles of stones at the narrow sides. The alignment between these piles of stones shows the rise of the Pleiades around 610 AD, which G. Hawkins also confirmed, and he therefore spoke of the "Plaza de las Pleiades".

The figure of the spider is located north of the large rectangular area. There is a very long line that, in one line center, begins east of the spider and runs through the spider towards the west, pointing to the sinking of the star Rigel in Orion around 100 BC. This line is also the connection between the two line centers.

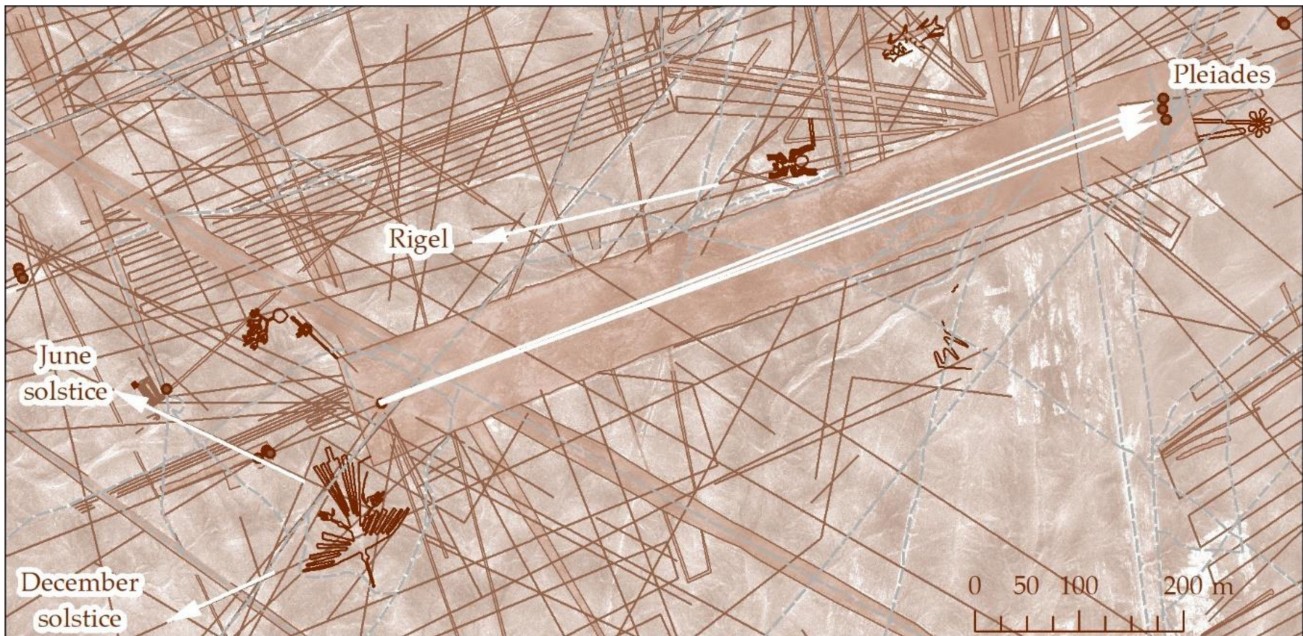

**Figure 10.** Stone heaps and lines with an astronomical alignment according to Maria Reiche [24].

Figure 11a shows the figure of the heron, which can be found in the east of the Pan-American Highway, approximately one kilometer northeast of the Pleiades area. The figure is approximately 300 m long and the 185 m long beak points exactly in the direction of the sunrise during June solstice. Much further north, in the Pampa of Palpa, there is a line about 3.9 km long and around 3 m wide, which Maria Reiche called the "Sirius Line" (Figure 11b) because it pointed to the sinking of Sirius, the brightest star, around 222 AC. The Sirius Line was badly damaged by the construction of the Pan-American Highway and a chicken farm. Strictly speaking, the Sirius line is not a line but a narrow, very long area. The approximately 3 m-wide area begins on a small hill in the east and ends in a narrow, around 600 m-long triangular area west of the Pan-American Highway.

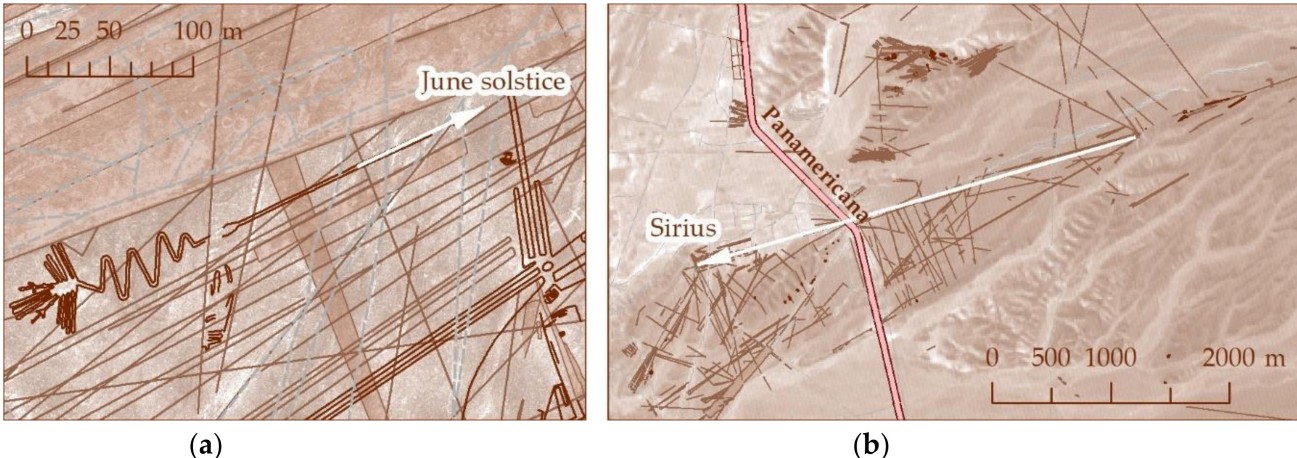

(**a**)　　　　　　　　　　　　　　　(**b**)

**Figure 11.** Two other orientations were documented by Maria Reiche: (**a**) the heron's beak points to the sunrise at the June solstice; (**b**) the so-called Sirius line is oriented to the sinking of Sirius during the Nasca time, around 222 AC.

Overall, it can be said that all of the checked geoglyphs identified by Maria Reiche point to the celestial bodies indicated by her [26]. However, this does not yet prove that the Nasca Pampa was actually a calendar system. It only shows that there are astronomically relevant geoglyphs.

### 3.3. Statistical Studies on the Astronomical Alignment of the Geoglyphs

3.3.1. Preview

As already mentioned, different types of geoglyphs can be found in the Pampa of Nasca and Palpa. The most common are lines and areas, and in addition, there are figures, stone formations and line centers. Examples of these different types of geoglyphs are visible in Figure 12. As already shown, the lines and areas are particularly relevant for astronomical investigations. However, there are also figures that have an astronomical orientation. For a first statistically reliable investigation, only the lines were selected. It must be considered that lines with a width of more than 0.5 m were recorded as areas in the NascaGIS (see Figure 12). Another point is that there are different types of lines. The NascaGIS distinguishes between the following types: straight, curved, meandering and zigzagging lines. Some examples are visible in Figure 12.

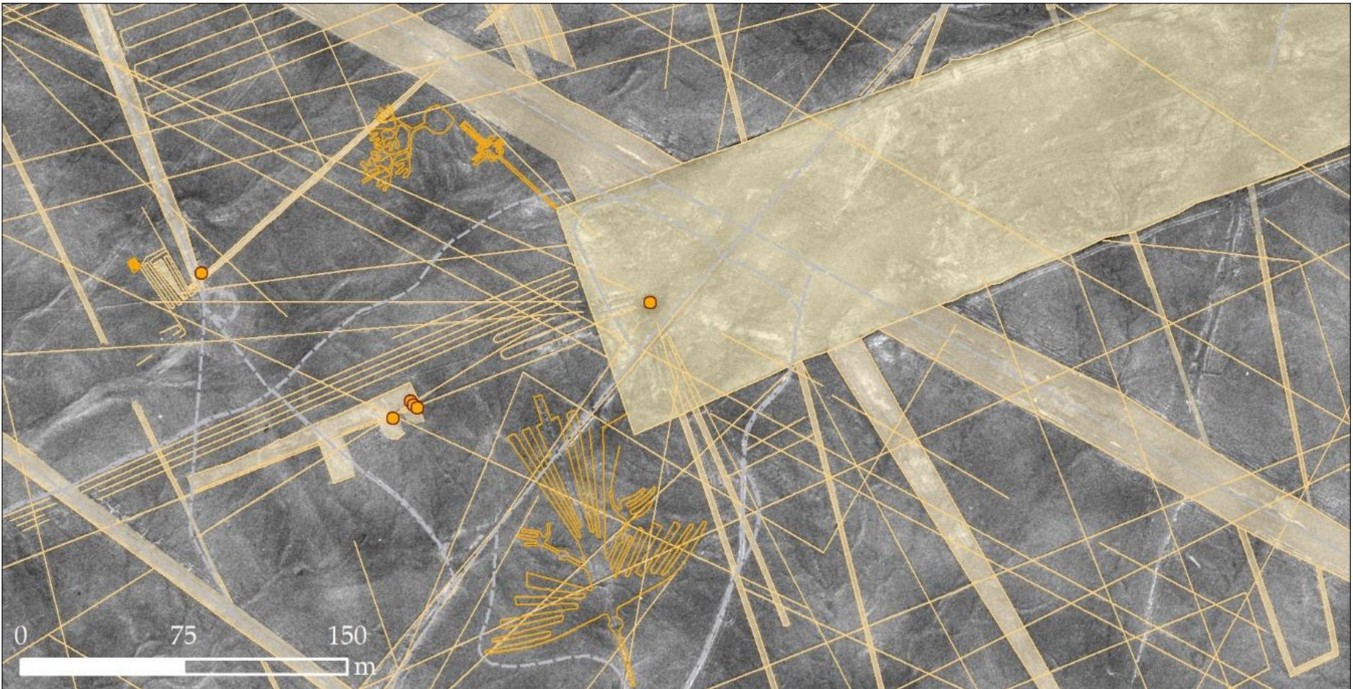

**Figure 12.** Extract from the NascaGIS: lines, areas, stone heaps and figures.

The first aim was a fully semi-automatic and systematic analysis with all straight lines, which are stored in the NascaGIS, to provide a statistical statement on the astronomical alignment of these lines. A total of 2308 Nasca Lines are of the type "straight line" and only these were initially included in the calculations. In the NascaGIS, all coordinates are stored in the PSAD 56 UTM Zone 18S system. From the coordinates of the start point A and end points B of each line, the two geodetic azimuths from A to B and from B to A were calculated. They differ by approximately 180 degrees; deflections of the vertical were neglected. The elevation angles were obtained with the help of digital terrain models.

Due to the atmosphere and the shape of the Earth, elevation angle corrections were necessary. The refraction as well as the curvature of the Earth was required. The point of extinction of a celestial body is more difficult to estimate. This effect depends on the Earth's atmosphere and the brightness of the object. There are very different statements in the literature: Müller [27] (p. 61) takes a value of 2 degrees, but there are examples that provide completely different values. Basically, the lighter a celestial body is, the longer it is visible. In these calculations, it was assumed that the Sun was always visible down to the horizon during the Nasca period.

Another problem is that nobody knows how the Nasca people used the sky and how competent they were at observing the celestial bodies. These questions cannot be answered because there are no written records. For this reason, the lines were examined for an astronomical signification in relation to the Sun and some brighter stars.

3.3.2. Investigation of the Correlation of Straight Lines with the Sun

All of the 2308 straight lines were used for the calculations. However, the structure of these lines is different. The length of the lines varies from a few meters to several kilometers and, very often, long lines are interrupted by destruction. The restriction to the line type "straight lines" means that figures, areas, zigzags and other types of lines were initially not considered; this will be the subject of a further separate investigation.

The calculation of the azimuths of the rising and setting Sun in the Nasca period was carried out for respective appearances or phenomena of solstices and equinoxes as well as for the zenith passage. The anti-zenith was not considered. The center of the Sun served as the target point.

In order to obtain a general overview of the frequency of certain azimuths of the Nasca Lines, a histogram was calculated in an interval of 10 degrees (see Figure 13).

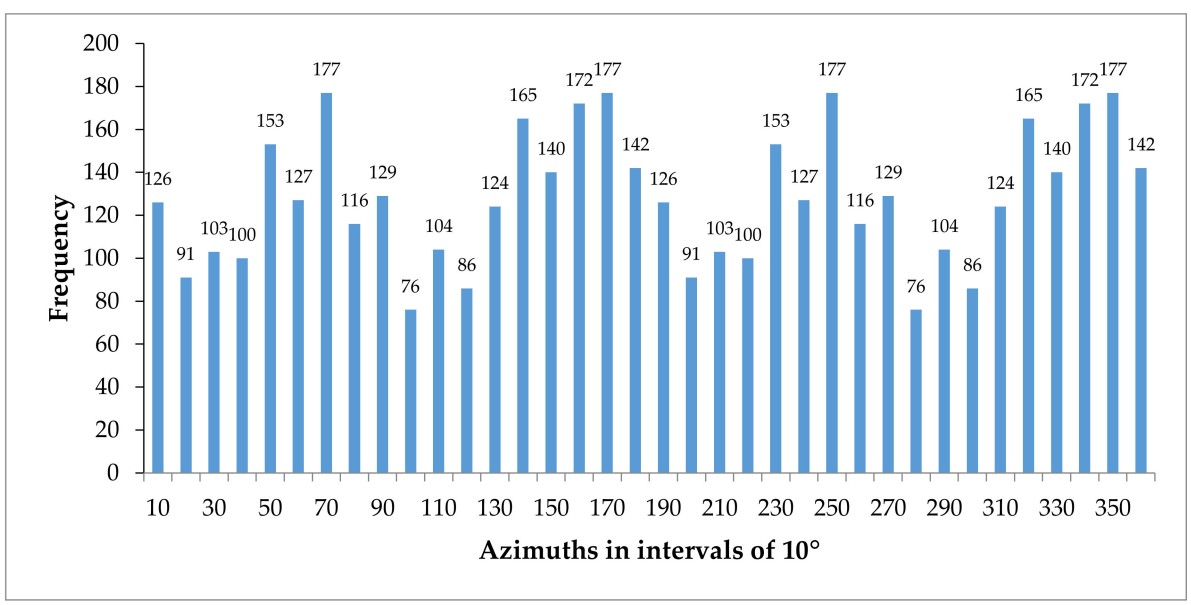

**Figure 13.** Frequency distribution of the azimuths of the Nasca Lines in intervals of 10 degrees.

This histogram clearly shows one symmetry because the opposite direction of the lines (azimuths differ by 180°) has exactly the same values for their frequency.

Within one year, the relevant azimuths in the Nasca region (mean latitude: −14.7 degrees) for sunrise are between approximately 65.6° and 114.4° and for sunset, they are between 245.6° and 294.4°; the histogram shows no noticeable frequency at these azimuths. Only 569 of the total 2308 straight lines fall into these windows.

In Table 2 are the numbers of lines which directly correspond to the exact azimuths (±2°) at the solstices and equinoxes as well as the zenith passages.

**Table 2.** Number of all line azimuths with reference to the Sun.

| Azimuths in Direction of | June Solstice | Equinox | December Solstice | Zenith | Total |
|---|---|---|---|---|---|
| Sunrise/Sunset | 43/19 | 24/24 | 19/43 | 8/5 | 94/91 |

The lines oriented to the sunrise at the equinox are exactly the same as the sunset lines (overall 24 lines). Further, the lines oriented to the sunrise in June solstice are the same ones for the sunset during the December solstice and vice versa (all together 62 lines). The zenith passages (13 lines) in particular seem to be very random. Regardless of this, it means that only 99 lines have an astronomical orientation to the Sun on those events. This corresponds to around 4% of all lines of the type "straight line".

### 3.3.3. Investigation of the Correlation of Straight Lines with Stars

For the investigation of the correlation of lines with stars, as a prerequisite, it would be necessary to identify, first, the celestial objects from which it can be assumed that they had a certain importance for the Nasca culture. Since there are no conclusive statements on this, the following procedure was chosen. The selection of stars to be examined was arbitrarily limited to 7 stars with a magnitude of less than 2. Sirius as the brightest star in the sky and Rigel as an important star in Maria Reiche's investigations were deliberately included. The other five (Achernar, Alkaid, Antares, Gacrux, Murzim) were chosen at random.

In addition to the brightness, the movement of the stars must also be considered. All stars move through space, although their distances from Earth are too great for the human eye to perceive their movements. Hence, the stars were once called "fixed stars" because they seemed to remain stationary in relation to one another; therefore, the stellar constellations remain nearly unchanged. However, today, we know that the constellations change their shape, albeit very slowly, because each star has an independent proper motion. These proper motions are essential parts of all well-known star catalogs such as the Fundamental Catalog 5 (FK5) and they are measured in an instantaneous coordinate system.

Another aspect to consider is the precession of the Earth. The rotation axis of the Earth describes, over a period of approximately 26,000 years, a small circle among the stars, centered on the ecliptic north pole and with an angular radius of about 23.5 degrees, an angle known as the obliquity of the ecliptic (see Figure 14).

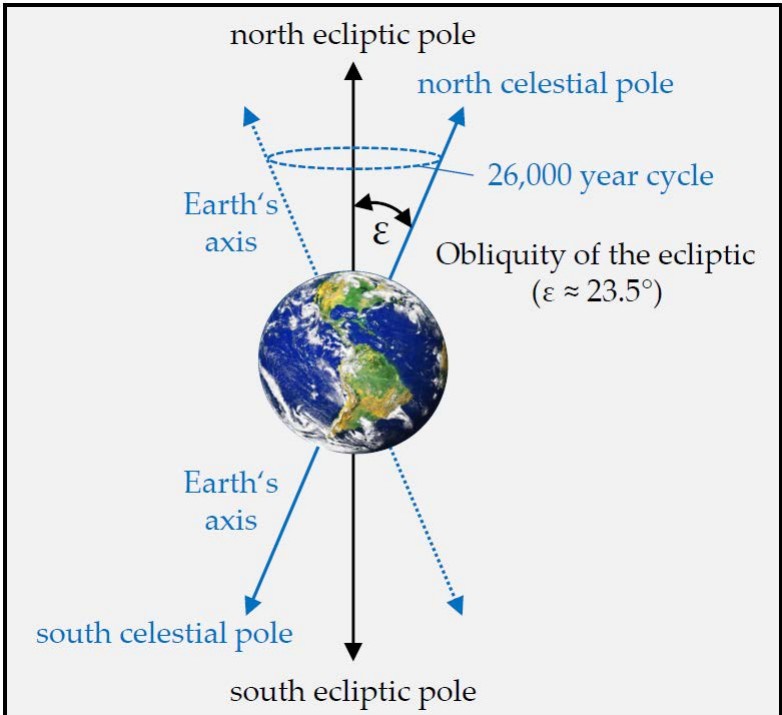

**Figure 14.** Precession—a gravity-induced, slow and continuous change in the orientation of the Earth's rotational axis in a cycle of approximately 26,000 years.

As a result, the equatorial coordinates $\alpha$ and $\delta$ (compare Figure 4a) of the stars must change with time (period of 26,000 years). For astronomical observations, the azimuth of the celestial bodies will change within the 1000 years of the Nasca time as shown in Table 3.

**Table 3.** Azimuth changes for selected bright stars near the horizon at Nasca.

| Year | Achernar | Alkaid | Antares | Gacrux | Murzim | Rigel | Sirius |
|---|---|---|---|---|---|---|---|
| −200 | 194.43° | 334.48° | 250.40° | 223.00° | 249.83° | 256.13° | 253.17° |
| 800 | 201.82° | 328.22° | 246.08° | 217.00° | 251.14° | 259.17° | 253.55° |
| Change | 7°24′ | 6°16′ | 4°19′ | 6°00′ | 1°19′ | 3°02′ | 0°23′ |

The changes in the azimuth are between 23 arcminutes and 7.4 degrees. Regardless of the length and nature of the lines, the following hits resulted (Figure 15):

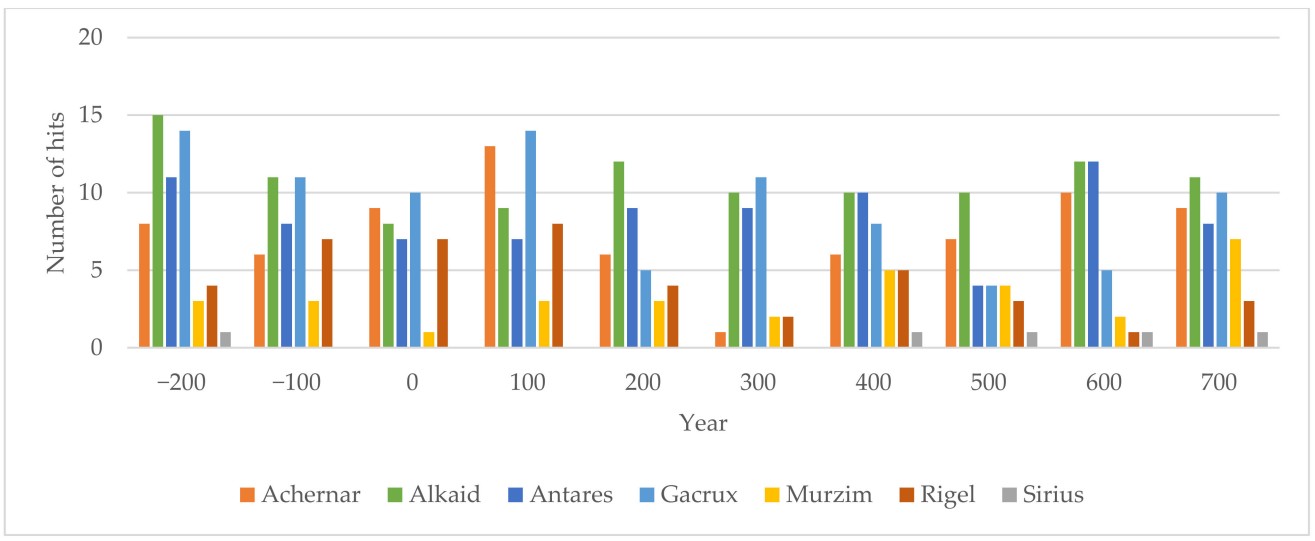

**Figure 15.** Total number of hits (at intervals of 100 years) for the correlation of straight lines with randomly selected bright stars during the Nasca period.

One result is, for the moment, very surprising. The brightest star on the firmament (Sirius) has the fewest hits. However, the comparison with Table 3 shows that the number of hits is directly dependent on the extent of the azimuth changes. Therefore, there does not seem to be a statistically relevant number of correlations between the straight lines and the stars. In addition, there is no accumulation of hits in a certain time interval. The hits found are distributed over the entire Nasca period. Therefore, the purely geometric calculations show a rather negative result for the straight lines with regard to the calendar theory.

However, the most important unknown, the atmospheric conditions during the Nasca period, can significantly change the result. First of all, the refraction should be mentioned here; the correction can reach approximately 35 arc minutes near the horizon. This applies equally to the Sun and the stars. For a very precise investigation, the refraction has to be studied and determined by long-term observations at the Pampa of Nasca. The extinction of the stars is even more problematic. In the Earth's atmosphere, it depends on the location, the wavelength and the elevation angle. If a star or another celestial body is close to the horizon, the path through the atmosphere is longer and the light attenuation is greater. The atmospheric extinction increases with the zenith distance. When stars disappear on the horizon mainly depends on the state of the atmosphere at the time of observation, but this is completely unknown [25].

## 4. Results

Based on the literature, it could be proven that the observation of the movement of the celestial bodies, especially the Sun, was of great importance for the pre-Columbian cultures. Numerous structures, and also geoglyphs, were aligned with the Sun. Maria Reiche suspected that the course of the Sun also played a special role for the Nascas. She assumed that the Nasca shamans used the geoglyphs to determine the annual calendar of agricultural and ceremonial activities, for example, to predict the times for sowing and harvesting. She also suspected that the figures at the pampa represent the stellar constellations of the Nasca culture.

During her more than 40 years of research, Maria Reiche found numerous lines, surfaces and figures with an orientation towards the Sun, several stars and the Moon. The examination of the geoglyphs found by Maria Reiche with an astronomical orientation towards the Sun and prominent stars during the Nasca period has been verified within this project.

In order to prove the calendar theory of Maria Reiche, however, a statistically reliable examination of all geoglyphs is necessary. Therefore, in a first step, all lines with a width of up to 50 cm and of the type "straight line" were examined with regard to their alignment with the Sun and prominent, very bright stars.

First of all, it was found that of these 2308 lines, only 569 point in the direction of any sunrise or sunset and of these, only 99 lines have a correlation with solstices, equinoxes or zenith passages. This corresponds to around 4% of all lines of the type "straight line". If one assumes that not only the solstices and equinoxes were of interest to the Nasca people, but also important seasonal events such as the beginning of the rainy season in the Andes, then other Sun positions could have been of interest too. Therefore, more precise agricultural knowledge must be included in the investigations.

Even with the stars, the result is not very satisfactory. Thus, over the entire period of the Nasca culture, there are always hits, i.e., lines that point to the selected very bright stars, but there is no accumulation in a certain period. It is also interesting that Sirius, the brightest star at the firmament, has the fewest hits. Hence, the number of hits does not allow any conclusions whether the hits are actually astronomically relevant lines or whether they are purely random.

## 5. Discussion and Conclusions

The results suggest that the geoglyphs, as a whole, certainly cannot be interpreted as a calendar system. Nevertheless, the astronomical theory cannot be completely negated. Therefore, in the next step, the other types of lines and areas should be examined more closely. Extensive manual preparations are required for this. There are, for example, very long meandering lines. These must be broken down into individual line sections in order to be able to calculate the astronomical azimuth of each line section. The same applies to the other line types. It must be defined for each individual area whether the calculations should be carried out for the center line or for the edges of the area. Another item are the figures and piles of stones. Relevant alignments to the Sun have already been demonstrated here as well. In order to be able to make a certain relevant statement, further investigations must also be carried out for these items.

One topic that has not yet been considered are geoglyphs, which have an orientation to the Moon. Maria Reiche defined a relatively large number of lines as so-called moon lines. The review of these lines is still pending, but the corresponding software programs are already available.

However, referring to the other pre-Columbian cultures, the orientation of geoglyphs towards the Sun seems the most likely and most promising. Therefore, these investigations will be in the foreground in the near future.

All geoglyphs that have an astronomical orientation are provided with the appropriate attributes in NascaGIS. The coordinates, azimuths and elevation angles used for the calculation as well as the correlating celestial bodies with the corresponding time information

are recorded in the tables of the NascaGIS data model [16]. In the future, all data should be available in the WebGIS application. At the moment, only the vector data can be called up as Web Map Service (WMS) [28], since many attributes in NascaGIS are still incomplete.

**Author Contributions:** Conceptualization, B.T. and C.R.; methodology, B.T.; formal analysis, B.T. and C.R.; investigation, K.P., B.T. and C.R.; writing—original draft preparation, K.P., B.T. and C.R.; writing—review and editing, K.P., B.T. and C.R.; visualization, C.R.; supervision, B.T.; project administration, B.T. All authors have read and agreed to the published version of the manuscript.

**Funding:** This research was funded by the Czech Technical University in Prague Grant number SGS20/053/OHK1/1T/11, and the APC was funded by the Czech Technical University in Prague, FCE, dept. of Geomatics".

**Institutional Review Board Statement:** Not applicable.

**Informed Consent Statement:** Not applicable.

**Acknowledgments:** The authors would like to thank the University of Applied Sciences (HTW), Faculty of Spatial Information, and the Faculty of Civil Engineering, CTU in Prague, for support.

**Conflicts of Interest:** The authors declare no conflict of interest. The funders had no role in the design of the study; in the collection, analyses, or interpretation of data; in the writing of the manuscript, or in the decision to publish the results.

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
