# Peer review of "Astronomical Investigation to Verify the Calendar Theory of the Nasca Lines"

_applsci, doi:10.3390/app11041637_

Round 1

Reviewer 1 Report

This is a interesting paper that takes a refreshingly sober look at a very evocative subject likely to be of interest to the general public. The paper usefully confirms Reiche's earlier observations about certain celestial alignments, and then critically examines a hypothesis that the Nasca lines were a calendar of sorts. The discussion of the astroarchaeological principles that underly the study are sound. The analysis appears to have been done rigourously and the results seem convincing -- there are many many lines with no apparent orientation to major celestial events. Lunar alignments are not considered although this next step has been preregistered here.

The results are however not strictly speaking reproducible as it stands; raw data on the orientation of the lines are not given (perhaps the NascaGIS is available but if so this should be cited) nor are the detailed analytical steps described, or than the functioning of software developed 'Within the scope of many diploma theses' (L295).

To redress this, think these theses should be cited in the bibliography, and the readers informed of where / when the full set of survey data will be available.  

Author Response

Dear reviewer, thank you for your comments.

The grammar was corrected by a native speaker from the USA.

The diploma theses on the astronomical calculations were added as references in line 299 and in the list of references: line 553-562.

The information on the NascaGIS was added in lines 492-497 and in the references line 573-574.

Reviewer 2 Report

I find paper subject quite interesting and recommend it for publishing.

In lines 71 and 377 there are some errors in brackets that should be dealt with.

Research was well conducted.

All the needed parameters were included.

It would be interesting to see further research on this topic.

Author Response

Dear reviewer, thank you for your comments.
The grammar was corrected by a native speaker from the USA.

There are no brackets on lines 71 and 377. But the figure reference from line 72 has been moved to the more appropriate place in line 71.

Reviewer 3 Report

The paper is very interesting given the number of theories about the purpose of Nazca lines. I definitely consider the use of statistical analysis of GIS data to be beneficial, despite the fact that the conclusions of the article did not definitively confirm or rule out the astronomical orientation of the lines. Further research is clearly needed, as the authors themselves point out. However, this article is a much-needed contribution to the discussion on the purpose of the Nazca lines.
I have only a few minor comments on the article, or suggestions for further research:
- I understand the use of random selection of stars to analyze the potential orientation of the lines - at least the low number of hits for the star Sirius stood out more. However, it may be appropriate to use, for example, the 10 brightest stars and not only Sirius and Rigel. Of course, I don't insist on the addition, it would only be interesting to see the results.
- When detecting hits, it would be appropriate to consider the tolerance with which the line can fall into a given category of stars (eg about 1 °) - this could eliminate the low abundance of the stars Sirius and Rigel. Do we know with what precision the people of that time were able to make astronomical observations?
- I think it would be better to change the description of the X-axis in the figure 13 to 200BC-100BC, 100BC-0, 0-100AD, ... 700AD-800AD
- Just out of curiosity - did the authors try to perform a similar analysis for a more extended period of time? E.g. -1000 to + 1000 years? How many hits would be e.g. for 10,000 BC (eg to refute various conspiracy theories)?

Author Response

Dear reviewer, thank you for your comments.
The grammar was corrected by a native speaker from the USA.

Fig. 13 only gives a general overview of the frequency of all azimuths of the Nasca lines in an interval of 10 degrees. A temporal assignment cannot be made here.

Investigations outside of the Nasca period have not yet been carried out, as it is considered certain that the geoglyphs in the Nasca pampas are attributable to the Nasca culture. Archaeological artefacts in the pampas suggest this.

With regard to the tolerances, 1 ° was assumed for the sun and 0.5 ° for the stars. As far as we know, there is no precise statement about how exactly the Nascas were able to observe the stars.

The following are planned for future calculations: the moon and all visible stars with a magnitude <2